# Tooth Loss and Oral Health-Related Quality of Life: A Study in a Convenience Sample from Austria

**DOI:** 10.3390/dj13100475

**Published:** 2025-10-17

**Authors:** Ana Nikolic, Stefanie Schindler, Hanns Moshammer

**Affiliations:** 1Department of Environmental Health, Center for Public Health, Medical University of Vienna, Kinderspitalgasse 1, 1090 Vienna, Austriastefanie.schindler@ages.at (S.S.); 2Department for National Reference Centers, Institute for Medical Microbiology and Hygiene Vienna, AGES-Austrian Agency for Health and Food Safety, Währinger Straße 25a, 1090 Vienna, Austria

**Keywords:** oral health impact profile, OHIP, tooth loss, oral health-related quality of life, OHRQoL, Austria

## Abstract

**Background**: To examine the association between the number and location of missing teeth and oral health-related quality of life (OHRQoL) as well as self-assessed general health in a convenience sample of Austrian dental patients, and to evaluate the applicability of the OHIP-G 14 and OHIP-G 12 instruments in this context. **Methods**: An anonymous questionnaire was distributed to adult patients in the waiting area of a single Austrian dental practice. It included the German version of the OHIP-14, a visual analog scale (EQ-5D VAS) for self-assessed general health, and demographic questions. A dentist clinically assessed the number and location of missing teeth using the FDI tooth numbering system. Regression models were used to evaluate the impact of anterior and posterior tooth loss on both outcome measures. The role of tooth replacement was also explored. A sensitivity analysis was conducted using the OHIP-G 12, a modified version excluding two items from the OHIP-14. **Results**: OHRQoL was significantly associated with the number of missing anterior teeth, while self-assessed general health was associated only with missing posterior teeth. These associations were consistent across models using OHIP-G 14 and OHIP-G 12. Tooth replacement, as assessed, showed no significant influence on either outcome. Age initially correlated with both outcomes but lost significance after controlling for the number of missing teeth. **Conclusions**: The hypothesis that anterior tooth loss negatively affects OHRQoL was supported. The association between posterior tooth loss and general health perception was not expected and requires further investigation. The OHIP-G 12 proved to be a valid and efficient alternative in this setting.

## 1. Introduction

Although significant decline in the incidence and prevalence of severe tooth loss can be observed in recent decades [1], studies show that oral conditions—including tooth loss or conditions leading to tooth loss—remain a substantial population health challenge [2]. Therefore, the impact of tooth loss on health and nutritional status [3,4], the impact of (self-assessed) general health on losing teeth, as well as the impact of tooth loss on OHRQoL, is of high concern. Previous research has consistently shown that tooth loss is associated with a reduction in oral health-related quality of life (OHRQoL). Individuals with missing teeth often report greater functional limitations, psychological discomfort, and social impairment, all of which contribute to lower OHRQoL scores [5,6]. This association is particularly evident when anterior teeth are missing, due to their relevance for both aesthetics and speech [7].

Oral health-related quality of life (OHRQoL) can be assessed through questionnaires. One very often used instrument is the Oral Health Impact Profile (OHIP). The original version proposed by Slade and Spencer [8] consists of 49 questions covering seven dimensions: functional limitations, pain, psychological discomfort, physical impairment, psychological impairment, social impairment, and disability. Slade himself [9] has later proposed and validated a shorter version consisting of only 14 questions. He concluded: “In a multivariate analysis of dentate people, eight oral status and sociodemographic variables were associated (*p* < 0.05) with both the OHIP-49 and the OHIP-14. While it will be important to replicate these findings in other populations, the findings suggest that the OHIP-14 has good reliability, validity and precision.”

The OHIP-14 has been translated into several languages and been widely applied in different settings and cultural contexts as well as for different health endpoints [10,11,12,13,14,15,16,17,18,19]. But Campos et al. [20] have pointed out that even when an instrument like the OHIP-14 is validated for a specific population, it does not mean necessarily that it is also valid in every other population group with different cultural background or health status. For example, a Portuguese version of the OHIP-14 has been developed and tested in the Brazilian context [21] but still Alfonso et al. [22] felt the need to test the instrument also in Portugal. A German version of OHIP has also been developed and were even used as a kind of gold standard in assessing other instruments [23]. Indeed, MacEntee and Brondani [24] acclaim the German version (OHIP-G 49, [25]) as one of the few that “used all eight steps to claim cultural equivalence with the original OHIP”.

To the best of the author’s knowledge, when the current study was conceived, no study had yet used and evaluated the (German) OHIP in Austria. Assessing the applicability of a German-language questionnaire in Austria is pertinent, as Austrian German, though mutually intelligible with Standard German, exhibits distinct linguistic features that may influence the interpretation and effectiveness of the questionnaire, comparable to the differences between British and American English. Therefore, the initial main purpose of that study was to test the applicability and plausibility of the instrument in the Austrian context. In the meantime, two studies from Vienna [26,27] have successfully applied the OHIP in Austria, both the original version of the shorter OHIP-G 5 and an adapted OHIP-G 5 for schoolchildren. Additionally, Omara et al. [28] had proposed a slightly shorted (OHIP 12), omitting two items of the OHIP-14 and revising the scoring system. Indeed, given the affiliations of most of the authors of that paper, it is possible that the participants of that study also came from Austria, but this is not stated clearly in the paper [28].

Following these developments, the present study was conducted to investigate, within an Austrian convenience sample, the association between tooth loss—particularly anterior tooth loss—and oral health-related quality of life. The underlying hypothesis was that tooth loss—particularly the loss of anterior teeth—negatively affects OHRQoL, as measured by OHIP-G 14 and OHIP-G 12.

## 2. Materials and Methods

An anonymous questionnaire-based survey was carried out in the waiting area of a single Austrian dental practice as part of a diploma thesis [29]. Patient recruitment and data collection were conducted over the period from May 2023 to July 2023. The questionnaire contained the 14 German questions of the OHIP 14 mainly. Since age, sex and general health were considered possible confounders; also questions on sex and age group starting from 18 years (18–29, 30–39, 40–49, etc.) were included. General health was evaluated using the Visual Analog Scale (VAS) of the EQ-5D-5L [30], as it provides a simple, validated, and widely accepted method for capturing overall health perception. Participants rated their current general health on a scale from 0 to 100, with 100 representing the best imaginable health and 0 the worst. The OHIP questions allow for five answers from “never” (0) to “very often” (4) with higher scores indicating poorer health.

Inclusion criteria comprised individuals aged 18 years or older who were present in the dental practice during the recruitment period and agreed to participate in the survey. A convenience sample was used, with participants being randomly selected in the waiting area, as this method proved to be both practical and time-efficient in the context of a routine dental care environment. Participants were recruited regardless of the reason for their dental visit or whether they had missing teeth, as the aim was to include a broad cross-section of adult dental patients. Individuals were excluded if they were under 18 years of age or if they self-reported insufficient proficiency in the German language, which could impede comprehension of the questionnaire.

Besides the OHIP 14 score, based on the findings of Omara et al. [28], it was decided to apply and analyze the score of the simplified version of the OHIP-12 in the study by excluding items 6 and 10. This additional scoring approach was included to allow for comparability with other studies and to assess the robustness of the findings across different versions of the OHIP instrument. The total scores (OHIP 14 and OHIP 12, respectively) were calculated by adding the scores of each single question. OHIP 14, thus, could take a score between 0 (best oral health) and 56 (worst oral health), OHIP 12, thus, could take a score between 0 (best oral health) and 48 (worst oral health).

A formal pretest was not conducted, as the study used a validated questionnaire, which has already been extensively tested for reliability and comprehensibility. In addition, the questionnaire was administered in a clinical setting where participants could ask for clarification if needed.

Additionally, a clinical dental examination was performed by a licensed dentist to assess and document the number and location of missing teeth. The assessment included a visual inspection and dental charting of all teeth, with a specific distinction between anterior teeth (incisors [FDI 11–12, 21–22, 31–32, 41–42] and canines [FDI 13, 23, 33, 43]) and posterior teeth (premolars [FDI 14–15, 24–25, 34–35, 44–45] and molars [FDI 16–18, 26–28, 36–38, 46–48]). A missing tooth was defined as any natural tooth that was no longer present, regardless of whether it was prosthetically replaced or not.

The data were first entered into a Microsoft Excel sheet and subsequently imported into STATA [31] for analysis. Descriptive statistics were calculated for all variables. Associations between the number of missing teeth and OHIP-G14 scores were assessed using linear regression analysis, with models estimated for the total number of missing teeth as well as for missing anterior and posterior teeth separately. As part of a sensitivity analysis, OHIP-14 was replaced with OHIP-12 in otherwise identical regression models. Additionally, a binary variable indicating whether missing teeth had been replaced was created. Participants with both replaced and non-replaced missing teeth were excluded from this analysis, as they could not be clearly classified into either group, which would have introduced ambiguity into the interpretation of the binary replacement variable. The “replaced” variable was included either as an additional predictor or as part of an interaction term with the number of missing teeth (anterior/posterior).

The effect size for the influence of missing teeth on the average OHIP-G 14 score was used analogously to Anbarserri et al. [32] who found an effect size of f^2^ = 0.25. This corresponds to a medium effect size. Since the hypothesis was specifically formulated, one-sided testing was carried out. Assuming a probability of error of 5% and a test power of 80%, a sample size calculation showed that 67 patients must be included. For safety reasons and to counteract the risk of people dropping out of the study, it was planned to survey 80 patients.

During the preparation of this work, the authors used ChatGPT (GPT-4-turbo, June 2024 version) to support various aspects of the writing process, including translation from German to English, linguistic refinement, structural clarity, and phrasing suggestions. All content generated with the assistance of this tool was critically reviewed and edited by the authors, who take full responsibility for the final version of the manuscript.

## 3. Results

Participants generally had no difficulty answering the OHIP questionnaire as indicated by subjective assessments and occasional feedback, as well as spot checks of participants during completion and the completeness of the filled-out questionnaires.

In the end, there were 83 participants (34 male and 49 female). The sexes did not differ significantly by age group (Fisher’s exact test: *p* = 0.421). The distribution of missing teeth showed a bimodal tendency with most of the participants (71%) having no missing anterior teeth, 8 participants (approximately 10%) having no missing teeth at all. Among those with missing teeth most had fewer than 10 missing teeth, with a concentration in the range of 4 to 6 and a few individuals showing extensive tooth loss (Figure 1). The descriptive statistics for the total sample and per sex are provided in Table 1.

OHRQoL according to OHIP 14 was not affected by sex and declined by age-group, both in the total sample and in both sexes separately. The same was true for the self-assessed general health (Table 2). Therefore, age group had to be considered a confounder, but not sex.

Table 3 and Table 4 show the factors determining OHRQoL determined by OHIP 14 score and OHIP 12 score, respectively, while Table 5 shows factors determining self-assessed general health. Models included either total number of missing teeth or missing anterior teeth and missing posterior teeth separately. Separate models were also calculated for the total sample and for males and females, respectively. Since age group no longer remained significant after control for number of missing teeth, it was not included in the more complex models.

The analysis showed that both OHIP-14 and OHIP-12 yielded comparable results, with OHRQoL being more strongly associated with the number of missing anterior teeth, while self-assessed general health was associated with the number of missing posterior teeth.

The inclusion of the binary variable ‘replaced’ in the regression models—either as main effect or in interaction with missing teeth—did not lead to statistically significant changes in effect estimates. No statistically significant associations were observed).

## 4. Discussion

The use of the German version of the OHIP instrument, more specifically the OHIP-5, has meanwhile proven to be easy also in the Austrian context in other studies [26,27]. In the present study, using the OHIP-14, no difficulties were observed in this Austrian convenience sample, suggesting that the instrument is also applicable in this setting.

In this study population OHRQoL (measured with OHIP 14 or OHIP 12) was not affected by gender and by self-assessed general health. Both, the OHRQoL and the self-assessed general health, were affected by age, but age only displayed an indirect effect mediated through loss of teeth. Although both OHRQoL (measured with OHIP 14 or OHIP 12) and self-assessed general health are negatively affected by number of missing teeth, the two are not confounded by each other.

There was a strong correlation between OHRQoL (measured with OHIP 14 or OHIP 12) and missing teeth. This is not so surprising given the important role of the teeth for OHRQoL, particularly considering the difficulties tooth loss can cause with essential functions like chewing and speaking [5,6,7,31,32,33,34,35,36]. Also, the association with the loss of anterior teeth, in particular, has been described in previous studies [7,33] and highlights the importance of aesthetics, which, when impaired, can lead to psychological and social stress, ultimately affecting the quality of life. Although the strength of the observed association was greater than expected, the results overall confirm the working hypothesis that tooth loss is significantly associated with reduced oral health-related quality of life.

Unlike OHRQoL, which was associated with anterior tooth loss, self-assessed general health was negatively affected by the number of missing posterior teeth. While the link between anterior tooth loss and reduced OHRQoL is well established, the similarly strong association between posterior tooth loss and self-assessed general health was not anticipated and was not based on a predefined hypothesis. One possible explanation is that loss of posterior teeth impairs mastication, which may influence nutrition and, consequently, perceived general health, but this association has not been widely addressed in previous studies, it may represent a clinically relevant link that warrants further investigation.

It is worth noting that in this study, self-assessed general health was originally included only as a potential confounder, given that several chronic diseases are known to increase the risk of tooth loss [37,38,39,40,41,42,43] and may also negatively affect OHRQoL. Indeed, the association can go both ways, because diseases including mental diseases [44,45,46,47,48], epilepsy [49], cancer (therapy) [50], or diabetes can lead to caries and loss of teeth [51], but chronic oral inflammation can also have severe systemic consequences [52] while tooth loss can affect nutrition [53]. Therefore, this study only used a very simple instrument to measure general health by self-assessment. The self-evaluation, while useful, has several limitations, including subjectivity, the potential for inaccurate self-perception, social desirability bias, and the influence of temporary factors that may not reflect long-term health status. Additionally, reverse causation of this association should be considered [43] and can never be ruled out in a cross-sectional study. Nevertheless, the association between number of missing posterior teeth and self-assessed general health was noteworthy and should be studied in more detail.

This study was conducted in a convenience sample of patients of a single dentist. Therefore, the sample is not representative of the Austrian population and the external validity of the results is limited. But if the association between number of missing (posterior) teeth and (self-assessed) general health was to be causal in this sample, there are no plausible reasons why this causality should be specific to this single sample.

Previous studies have shown that restoring missing teeth with a dental prosthesis has a huge impact on the OHRQoL. While some studies see this independent of the analyzed type of dental prosthetic restoration [54], others highlight the differences [55,56]. In this study, no significant associations were found for the replacement of missing teeth, which may reflect limitations in how the variable was defined or the restricted sample size. Additionally, the type of dental prosthesis in detail was not analyzed, which may have been a relevant factor influencing the outcomes.

Another limitation of the study is the omission of socioeconomic factors, which are known potential confounders and have been identified as such in previous research [7,33]. Socioeconomic status may influence both the number of missing teeth [57] and oral health-related quality of life (OHRQoL) [58,59] and thus represents a relevant variable in this context. Socioeconomic status is also associated with quality of life in general and with disease risk [60,61,62,63] and chronic disease can have negative consequences for socioeconomic status [64]. However, collecting data on socioeconomic status was considered problematic, as such questions might be perceived as sensitive and could have negatively impacted the response rate. Therefore, these factors were not assessed.

## 5. Conclusions

This study confirmed that tooth loss—particularly of anterior teeth—is significantly associated with reduced oral health-related quality of life, as measured by both OHIP-G 14 and OHIP-G 12. In contrast, self-assessed general health was associated only with the number of missing posterior teeth. Tooth replacement, as assessed in this study, showed no significant effect. These findings underline the importance of preserving anterior teeth for maintaining OHRQoL. However, due to the limited sample size and the setting in a single Austrian dental practice, the results are not generalizable and should be interpreted with caution. Future studies with larger and more diverse populations are needed to validate and expand upon these findings.

## Figures and Tables

**Figure 1 dentistry-13-00475-f001:**
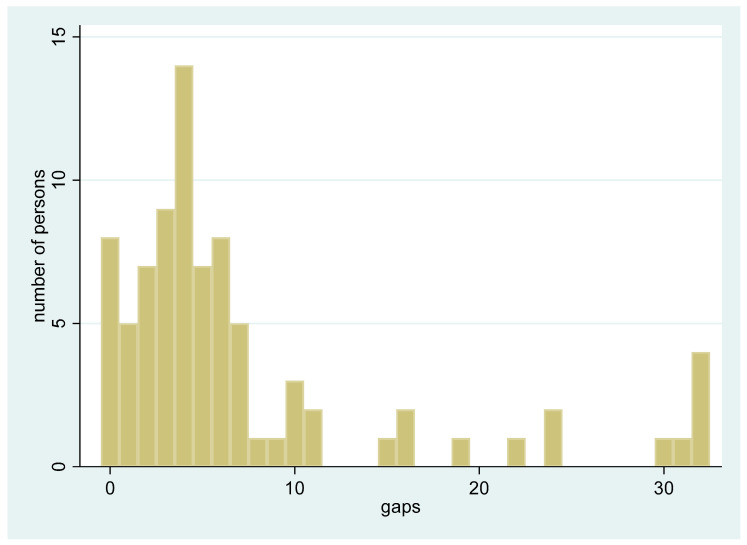
Distribution of the total number of missing teeth among all participants (*n* = 83). The *x*-axis represents the number of missing teeth (gaps) per individual (from 0 to 32), while the *y*-axis shows the relative frequency (proportion) of participants with each corresponding number of missing teeth.

**Table 1 dentistry-13-00475-t001:** Description of the sample.

Parameter	All (*n* = 83)	Male (*n* = 34)	Female (*n* = 49)
Age group			
18–29	25	11	14
30–39	19	5	14
40–49	15	8	7
50–59	15	7	8
60–69	4	2	2
70–79	1	0	1
80–89	3	0	3
90 and more	1	1	0
Participants without missing anterior teeth (*n*, %)	59 (71%)	24 (71%)	35 (71%)
Participants without missing posterior teeth (*n*, %)	11 (13%)	5 (15%)	6 (12%)
Mean number of missing anterior teeth ^1^ (SD)	5.8 (4.4)	5.6 (4.2)	5.9 (4.7)
Mean number of missing posterior teeth ^1^ (SD)	6.6 (5.5)	6.6 (5.1)	6.6 (5.7)
Mean OHIP-14 score (SD)	12.0 (12.1)	11.3 (13.1)	11.8 (11.5)
Self-assessed general health, mean VAS score (SD)	82.3 (9.5)	81.4 (8.5)	83.0 (10.2)

^1^ Calculated only among participants with ≥1 missing teeth in the respective region.

**Table 2 dentistry-13-00475-t002:** Effect of age (and sex) on general health (GH, higher values indicate better health) and OHRQoL according to OHIP 14 (lower values indicate better health).

Parameter (Outcome)	Coefficient	*p*-Value *	95% Confidence Interval
**Factors affecting OHIP 14**			
Total sample			
Male (versus female)	0.54	0.828	(−4.39232; 5.47526)
Age group (linear)	**3.14**	**<0.001**	**(1.663605**; **4.618386)**
Stratified analysis, OHIP 14 in males			
Age group (linear)	**3.54**	**0.010**	**(0.8955936**; **6.186678)**
Stratified analysis, OHIP 14 in females			
Age group (linear)	**2.89**	**0.002**	**(1.089216**; **4.698753)**
**Factors affecting General health (GH)**			
Total sample			
Male (versus female)	−1.58	0.436	(−5.601429; 2.438769)
Age group (linear)	**−1.95**	**0.002**	**(−3.151901**; **−0.7443168)**
Stratified analysis, GH in males			
Age group (linear)	**−1.94**	**0.034**	**(−3.715767**; **−0.1575545)**
Stratified analysis, GH in females			
Age group (linear)	**−1.96**	**0.023**	**(−3.622198**; **−0.2881524)**

* Bold font indicates *p* < 0.05.

**Table 3 dentistry-13-00475-t003:** Factors determining OHRQoL measured with OHIP 14 (coefficient, *p*-value *).

Parameter (Model)	All	Male	Female
OHIP 14 model 1			
Age group (linear)	−0.66 (0.459)	0.07 (0.956)	−1.22 (0.307)
Total Missing teeth	**1.03 (<0.001)**	**1.09 (0.001)**	**1.03 (<0.001)**
OHIP 14 model 2			
Missing anterior teeth	**1.67 (<0.001)**	**1.94 (0.008)**	**1.50 (0.007)**
Missing posterior teeth	0.50 (0.061)	0.60 (0.179)	0.46 (0.177)
OHIP 14 model 3			
Missing anterior teeth	**1.71 (<0.001)**	**1.85 (0.013)**	**1.58 (0.004)**
Missing posterior teeth	0.43 (0.129)	0.72 (0.139)	0.27 (0.442)
Self-assessed general health	−0.08 (0.514)	0.15 (0.709)	−0.20 (0.149)

* Bold font indicates *p* < 0.05.

**Table 4 dentistry-13-00475-t004:** Factors determining OHRQoL measured with OHIP 12 (coefficient, *p*-value *).

Parameter (Model)	All	Male	Female
OHIP 12 model 1			
Total Missing teeth	**0.82 (<0.001)**	**0.95 (<0.001)**	**0.74 (<0.001)**
OHIP 12 model 2			
Missing anterior teeth	**1.44 (<0.001)**	**1.66 (0.007)**	**1.30 (0.007)**
Missing posterior teeth	0.45 (0.052)	0.54 (0.153)	0.41 (0.174)
OHIP 12 model 3			
Missing anterior teeth	**1.64 (<0.001)**	**1.71 (0.012)**	**1.57 (0.005)**
Missing posterior teeth	0.49 (0.076)	0.82 (0.067)	0.29 (0.417)
Self-assessed general health	−0.08 (0.499)	0.13 (0.509)	−0.19 (0.164)

* Bold font indicates *p* < 0.05.

**Table 5 dentistry-13-00475-t005:** Factors determining self-assessed general health (coefficient, *p*-value *).

Parameter (Model)	All	Male	Female
Self-assessed general health model 1			
Age group (linear)	−1.19 (0.173)	−1.81 (0.127)	−0.57 (0.660)
Total missing teeth	−0.20 (0.231)	−0.04 (0.859)	−0.35 (0.167)
Self-assessed general health model 2			
Missing anterior teeth	0.49 (0.226)	0.64 (0.267)	0.40 (0.484)
Missing posterior teeth	**−0.88 (0.001)**	**−0.81 (0.034)**	**−0.92 (0.013)**
Self-assessed general health model 3			
Missing anterior teeth	0.61 (0.172)	0.44 (0.501)	0.74 (0.227)
Missing posterior teeth	**−0.85 (0.002)**	**−0.88 (0.028)**	**−0.82 (0.028)**
OHIP 14	−0.07 (0.514)	0.10 (0.494)	−0.23 (0.149)

* Bold font indicates *p* < 0.05.

## Data Availability

The datasets generated and/or analyzed during the current study are available from the corresponding author on reasonable request.

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
