# Peer review of "Tooth Loss and Oral Health-Related Quality of Life: A Study in a Convenience Sample from Austria"

_dentistry, 2025, doi:10.3390/dj13100475_

Round 1

Reviewer 1 Report

Comments and Suggestions for Authors

With interest I’ve read the paper «Tooth Loss and Oral Health-Related Quality of Life: A Study in a Convenience Sample from Austria». The authors assessed the association between the number and location of missing teeth and oral health-related quality of life as well as self-assessed general health in a convenience sample of Austrian dental patients. They found that anterior tooth loss negatively affected OHRQoL as well as that  posterior tooth loss negatively affected general health perception. The study is interesting and relevant. There are some minor comments to be addressed.

Introduction is sufficient, I would only suggest to add some results of studies from other countries to illustrate that the country may have impact on the studied parameter (and confirming the need for the study in Austrian population).

Materials and methods

It is better to avoid any discussions and explanations in this section and focus on how the study was accomplished.

What about the approval of the ethics committee?  Did you collect written informed consent?

How was the sample size justified?

Results

Lines 151-152 How did you assess the significance of differences in age distribution between gender groups? It was not stated in the materials and methods section.

Table 2 It is not intuitively understandable, I suggest to rearrange the first column to clearly indicate what is compared (especially the part for males and females separately).

Table 5 should be placed in the results section too.

Discussion

More discussion is needed regarding the comparisons with other studies which also assessed the impact of missed teeth on QoL.

Author Response

Please, find our response in the attached file!

Reviewer 2 Report

Comments and Suggestions for Authors

The manuscript is well-structured, with a clear introduction, an appropriate methodology, and consistent results. The discussion is balanced, linking the findings to the existing literature and transparently acknowledging the main limitations, particularly the small sample size and the single-clinic recruitment. The associations identified between anterior tooth loss and OHRQoL, as well as the unexpected relationship between posterior tooth loss and self-assessed general health, are relevant and merit attention.

I would suggest elaborating further on the absence of socioeconomic variables, given their well-documented impact in this field, even if the methodological rationale for their exclusion is understandable.

Overall, the manuscript is well-written, methodologically sound, and of interest to the scientific community.

Author Response

(The authors gave the same response as above.)

Reviewer 3 Report

Comments and Suggestions for Authors

Dear authors

I found the manuscript very well written. The fact that the sample is relatively small for the kind of statistic analyses you made has been noticed as a limitation.

As in Fig.1 you give the distribution of the missing teeth ranging 0-32 (that includes the 3rd molars), I would like you to specify if the missing 3rd mollars were counted in case of impacted or congenitally missing teeth. As the impact of these teeth on quality of life is smaller than the other teeth, an overestimation of the number of posterior missing teeth, especially in older persons, may affect the statistical results. I suggest you to repeat the statistical analyses,  excluding the 3rd molars.

I wish you the best

Author Response

(The authors gave the same response as above.)
